# Influence of Flour and Fat Type on Dough Rheology and Technological Characteristics of 3D-Printed Cookies

**DOI:** 10.3390/foods10010193

**Published:** 2021-01-19

**Authors:** Tomislava Vukušić Pavičić, Tomislava Grgić, Mia Ivanov, Dubravka Novotni, Zoran Herceg

**Affiliations:** Faculty of Food Technology and Biotechnology, University of Zagreb, Pierottijeva 6, 10000 Zagreb, Croatia; tvukusic@pbf.hr (T.V.P.); tomislavagrgic5@gmail.com (T.G.); mivanov@pbf.hr (M.I.); zherceg@pbf.hr (Z.H.)

**Keywords:** biscuits, extrusion rate, 3D printing precision and repeatability, dough consistency and viscosity, dietary fiber, carob flour, oat flour, olive oil

## Abstract

In this study, we designed high fiber cookie recipe without using additives by means of extrusion-based 3D printing. We aimed to relate printing quality and cookie physical properties with dough rheology and dietary fiber content depending on the flour (oat, rye, rice, and carob flour) and fat type (olive oil or butter). The flour choice influenced all cookie quality parameters: baking loss, color, line height and width, and dietary fiber content. Results indicated that lower baking loss and better printing quality were obtained for cookie dough containing olive oil, which had higher viscosity and consistency coefficient compared with dough containing butter. Cookies with olive oil in which part of the oat flour was replaced with rye and carob flour were printed with high accuracy (≥98%), close to the ideal 3D shape. Overall, this study demonstrates the importance of selecting fat and particularly flour, as well as the extrusion rate on the quality and repeatability of 3D-printed cookies.

## 1. Introduction

Recently, extrusion-based 3D printing methods became commonly used in the food sector, where a mixture or melted material is continuously pushed from a nozzle by pressure and deposited on printer platform to obtain the desired 3D shape [1]. Cereals materials intended for 3D printing should be homogenous and must have adequate extrudability to allow smooth extrusion and suitable viscosity/mechanical properties and to tolerate a vertical assembly of the 3D-printed structure during printing and postprocessing (e.g., baking) [2,3]. Wheat dough is almost an ideal material for 3D printing [4], but there are many challenges for researchers dealing with 3D printing of cereal foods, like how to obtain the maximum yield of selected homogeneous mixture that has a good 3D printability and the ability to create a stable structure after printing.

The rheological properties of the material are crucial for successful 3D food printing. A material for extrusion-based printing should display shear thinning behavior which is an indicator that the material can be extruded through the nozzle [5]. Physical properties such as relatively higher extrudability, gel strength, elasticity and relatively lower ductility have positive influence on the 3D shape of dough [6].

The composition of food has a more relevant effect on its rheological properties, and consequently on the retention of the printed structure than printing temperature [7]. Raw materials rich in fiber, protein, minerals, and vitamins should be carefully chosen to improve the rheological properties of the dough, as well as the quality and nutritional value of 3D-printed food [8]. Dietary fiber incorporated into food can impart its textural properties and material rigidity [9]. The addition of dietary fiber results in an increase water holding capacity, i.e., reducing the amount of unbound water compared to bound water and greater ability to bind water from the surface [9]. Unlike in conventional process, the processability of 3D cereal material might be enhanced with adding high-fiber ingredients. Still, the influence of various whole nonwheat and legume flours as a rich source of fiber and minerals on the rheological properties of 3D-printed dough needs to be explored.

The type and the amount of added fat has a strong effect on the desired rheological and textural properties of the dough [10]. The addition of triglycerides can effectively change the functional properties of 3D food pastes and end products depending on their composition and structure, including its melting point range, solid fat index and crystal structure [2]. Cookie recipes using wheat, rice, tapioca flour, and different types of fat (butter and shortening) were tested for the extrusion-based 3D printability and postprocessing capacity [11]. In previous research studies, butter was most often used source of fat. The influence of dough composition on the quality of 3D-printed food using wheat flour, freeze-dried mango powder, olive oil, and water was explored [1]. Yet, the difference between the effect of olive oil and butter on the quality of extrusion-based 3D printing and the final 3D structures was not compared previously.

The precision of 3D printing and the shape of final product are influenced by many technological factors: printing mechanism, material properties, printing process parameters, and postprocessing methods [12]. The accuracy of 3D-printed samples, in addition to the physical properties of the material, largely depends on the parameters of the printing process, including filament diameter, nozzle movement speed, nozzle diameter, and nozzle height [4]. The influence of processing parameters on the rheological properties and geometric accuracy of 3D-printed structures made of dough with low-gluten flour was examined [4]. Nevertheless, it is a challenge to find an optimum relationship between the rheological properties of dough and the process parameters, which is the key to improve the quality of the final 3D-printed structures.

In addition, 3D printable material must form stable structure with self-supporting properties of the deposited layers during and postprocessing without slumping, spreading, or bridging [5]. Postprocessing of cereal food involves the exposure of a printed 3D object to thermal processing during which chemical and physical changes occur, such as protein denaturation, starch gelatinization, moisture loss, changes in color, size, texture, and nutritional value of the product [13]. Controlling and modifying the recipe and adding additives are the main means to maintain the stability of 3D shapes during and after postprocessing treatments [14]. In previous research, the hydrocolloid addition on the dimensional stability of wheat cookies during temperature variations was proposed [15].

The main aim of this research was to investigate the effect of butter substitution with olive oil and the influence whole grain and carob flour on the rheological properties of the dough and finally on 3D printing accuracy and repeatability. In addition, the relationship between the printing process parameters and the rheological properties of the mixtures with the dimensions and accuracy of the 3D-printed forms was examined.

## 2. Materials and Methods

### 2.1. Materials

For experiments we used four types of flour, two types of fat and floral honey purchased at local grocery stores (Zagreb, Croatia). The used oat flour (Eko-Jazo Ltd., Ivanovac, Croatia) contained 72.1 g/100 g carbohydrates, 9.9 g/100 g proteins, 5.5 g/100 g lipids, and 2.3 g/100 g fiber. Rice flour (Garden Ltd., Zagreb, Croatia) had 79.6 g/100 g carbohydrates, 7.2 g/100 g protein, 2.2 g/100 g lipids, and 2.9 g/100 g fiber. Rye flour (Garden Ltd., Zagreb, Croatia) was composed of 77 g/100 g carbohydrates, 9 g/100 g protein, 0.74 g/100 g lipids, and 11 g/100 g fiber. Carob flour (Perna family-run farm, Vis, Croatia) contained 81.4 g/100 g carbohydrates, 5.3 g/100 g protein, 0.5 g/100 g lipids, and 26.3 g/100 g fiber. Butter contained 82 g/100 g lipids, whereas olive oil had 91 g/100 g lipids.

### 2.2. Preparation of Cookie Dough

To determine the dough recipe for 3D printing, using a slightly modified method according to Lipton et al. [14], the ratio of ingredients was varied until the dough with proper printing ability was obtained. Based on the preliminary result, the optimal ratio of dry matter and water suitable for extrusion-based 3D printing of dough was determined. According to Pulatsu et al. [11], with certain modifications of the recipe, six mixtures were prepared (Table 1).

### 2.3. The 3D Printing Process

The dough samples were fabricated using the extrusion-based 3D printer (Createbot 3D Food Printer-Multi-Ingredient Support, Ningbo Createbot Electronic Technology Co., Ltd. Zhejiang Province, Ningbo, China) with a syringe-extrusion-type plunger (volume of 20 mL). Before printing, the mixture was homogenized. The desired 3D shape of a flower (with 12 layers and layer height of 0.4 mm) was selected from software Cura 15.02.1. to examine the ability of dough to form accurate and stable structure. The 3D printing process was adjusted at temperature of 25 °C, print speed of 25 mm/s, nozzle diameter of 2.0 mm, and nozzle height of 2.0 mm.

### 2.4. Rheological Properties Measurements

The rheological properties of baking dough with different compositions were characterized using a RM 100 Plus Viscometer (Lamy Rheology Instruments, Champagne auMont d’Or, France) according to the Standard ISO 2555:2018 in mPas. The goal was to determine the rheological parameters at the same temperature. However, due to too high torsion during the determination of rheological parameters at 25 °C, measurements were performed at 30 °C. The temperature of samples was maintained at 30 ± 1 °C throughout the test using a Julabo F 33 refrigerated-heating circulator. The apparent viscosity (Pas) was measured at the shear rate 5 s^−1^. In addition, the consistency coefficient (Pas^n^) and flow behavior index were calculated.

### 2.5. Determination of Extrusion Rate

The extrusion rate for each 3D-printed cookie was determined according to Equation (1) [16], and it is reported as average value of 10 replicates.
(1)$$\mathrm{Extrusion}\;\mathrm{rate}\;\left(g/\min\right)=\frac{\mathrm{weight}\;\mathrm{of}\;\mathrm{printed}\;\mathrm{object}\;\left(g\right)}{\mathrm{printing}\;\mathrm{time}\;\left(\min\right)}.$$

### 2.6. Baking

Ten 3D shapes printed in parallels of the same mixture (5 cookies out of one syringe) were baked in a convection oven (Bistrot 664, BEST FOR, Italy), at 180 °C, for 3 min. The weight of the baked 3D-printed forms was determined after 30 min of cooling at ambient conditions using Sartorius A 120S analytical balance (reading 0.001 g). Baking loss was quantified by measuring the weight of the 3D-printed forms before and after baking according to the Equation below (2):(2)$$\mathrm{Baking}\;\mathrm{loss}=\;\frac{m_{1}-m_{2}}{m_{1}}\times 100.$$
where m_1_ is the weight of printed sample before and m_2_ after baking and cooling.

### 2.7. Analyses of Physical Properties of 3D-Printed Shapes

The weight of all printed shapes from each syringe filling was determined to calculate the utilization rate of the dough during 3D printing process for each mixture. The utilization rate indicates what percentage of total mixture input in 3D printer syringe entered into final product—cookie dough. Dough utilization rate was calculated as the ratio of the sum of weight of forms printed from one syringe to the initial weight of mixture inserted in the syringe.

The total height of baked cookies was determined at 4 positions (Figure 1a) for each sample using a DIGI-MET Digital Micrometer (Helios Preisser, Gammertingen, Germany) (Figure 2). The total height was divided by the number of layers (twelve) to calculate the line height. The width of the printed layer was evaluated with digital image analysis using the ImageJ software (National Institutes of Health). All baked 3D forms (10 replicates) were photographed (with a Nikon D7500, 48 mm lens, ISO 450-500, 1/60; f/5.6) from the same fixed distance and images were calibrated on a real scale. The width of the initial printed layer was estimated at 4 positions (Figure 1b and Figure 2) and is shown as mean value. According to the Huang [17], with a minor modification, $\sqrt{\frac{Q}{v}}$ was calculated for 10 shapes of each mixture using the Equation (3):(3)$$\sqrt{\frac{Q}{v}}=a/\sqrt{\frac{4}{\pi c}}$$
where Q—extrusion rate (mm^3^/s), v—print speed (mm/s), a—width (mm), and c—value defined as ratio of line height over width.

The line spreading (c-value) calculated as the line height/width ratio is later represented as the normalized value to ideal.

Since the printed and baked forms differed, the shape accuracy of each printed form from the desired one was quantified by analysis and comparison of binary photographs (ImageJ, National Institute of Health, Bethesda, MD, USA) of the finished shapes [18]. The last printed 3D shape, given its dimensions (height and width) was the most accurately printed, and was set as the desired shape. Using the ratio of the number of black pixels to the total number of pixels in the photo, the shape accuracy of individual printed objects in relation to the desired 3D object was quantified and expressed in percentage.

The color values of L^*^, a^*^, and b^*^ were determined on the bottom of the cookie sample where the area was the largest (Figure 1b) using Konica Minolta CM-700d spectrophotometer. The results are shown as the average ± standard deviation error of 10 samples of each mixture. The total color change ∆E and the parameters H° and C between the samples of mixtures A and B were calculated, according to the Equations (4)–(6):(4)$$\Delta E_{\mathrm{ab}}^{*}=\sqrt{{\left(L_{A}^{*}-{\;L}_{B}^{*}\right)}^{2}\;+\;{\left(a_{A}^{*}-{\;a}_{B}^{*}\right)}^{2}\;+\;{\left(b_{A}^{*}-{\;b}_{B}^{*}\right)}^{2}},\;$$
(5)$$H{}^{\circ}=\tan^{-1}\left(\frac{b^{*}}{a^{*}}\right),\;$$
(6)$$C=\sqrt{{a^{*}}^{2}+{b^{*}}^{2}},\;$$
where:$L_{A}^{*}-\mathrm{lightness}\;\mathrm{of}\;\mathrm{the}\;\mathrm{color}\;\mathrm{of}\;\mathrm{the}\;\mathrm{sample}\;\mathrm{of}\;\mathrm{mixture}\;A$$L_{B}^{*}-\mathrm{lightness}\;\mathrm{of}\;\mathrm{the}\;\mathrm{color}\;\mathrm{of}\;\mathrm{the}\;\mathrm{sample}\;\mathrm{of}\;\mathrm{mixture}\;B$$a_{A}^{*}-\mathrm{color}\;\mathrm{parameter}\;\mathrm{of}\;\mathrm{the}\;\mathrm{sample}\;\mathrm{of}\;\mathrm{mixture}\;A$$a_{B}^{*}-\;\mathrm{color}\;\mathrm{parameter}\;\mathrm{of}\;\mathrm{the}\;\mathrm{sample}\;\mathrm{of}\;\mathrm{mixture}\;B$$b_{A}^{*}-\mathrm{color}\;\mathrm{parameter}\;\mathrm{of}\;\mathrm{the}\;\mathrm{sample}\;\mathrm{of}\;\mathrm{mixture}\;A$$b_{B}^{*}-\mathrm{color}\;\mathrm{parameter}\;\mathrm{of}\;\mathrm{the}\;\mathrm{sample}\;\mathrm{of}\;\mathrm{mixture}\;B$

The value of C represents the saturation or intensity of the color, and the value of H° is the visual experience of the color (0–90° is the red-orange color, 90–180° is the yellow-green color, 180–270° is the blue-green color, and 270–360° is blue-purple).

### 2.8. Determination of Total Dietary Fiber and Moisture Content

The moisture content of cookies was determined according to AACC method 44-15.02 in triplicate. Total dietary fiber (TDF) of dried and defatted samples was determined according to AACC method 32-05.01 and AOAC Method 985.29 (if fat content is >10%) in duplicate using Total Dietary Fiber Assay Kit (Megazyme, Bray, Ireland). The results are expressed se mean value in grams per dry weight of cookie.

### 2.9. Statistical Analyses

The data recorded were subjected to one-way analysis of variance (ANOVA) to establish that there were no significant differences between two fillings of printer syringe with the same dough mixture. Two-way ANOVA was used to identify the effects of flour and fat type on sample parameters. ANOVA for repeated measures was applied on weight, line width, and height of 3D-printed shapes to evaluate the repeatability of printing. ANOVA, Tukey post hoc test, Pearson correlation test, and principal component analysis (PCA) were run with a Statistica 12 (StatSoft Inc., Tulsa, OK, USA). Differences were considered statistically significant when *p* < 0.05.

## 3. Results and Discussion

### 3.1. Rheological Properties of Dough

The rheological properties of the tested mixtures were significantly different depending on the flour and fat type (*p* < 0.001) (Table 2). The flow behavior index was less than one for all mixtures and the apparent viscosity decreased with the increased shear rate. This indicated that the cookie dough shows pseudoplastic, i.e., shear-thinning behavior. Due to its pseudoplastic behavior, dough can be easily extruded [8]. In addition, the extrusion process and the printability of the material largely depend on the consistency coefficient and the flow index [8]. It is noticeable that mixtures with olive oil (1A, 2A, and 3A) had a substantially higher viscosity and consistency coefficient than those with butter (1B, 2B, and 3B) (Table 2). This is in agreement with previous studies [8,10]. Jacob and Leelavathi [10] who studied the effect of four different types of fat (baking fat, margarine, emulsified fat produced to resemble butter, refined sunflower oil, and vegetable fat “dalda”) on the rheology of the cookie dough, stated that the dough containing oil is not only more viscous and more cohesive but also softer compared to doughs containing the remaining three fats. Martínez-Monzó et al. [7] examined the influence of mashed potato composition and temperature on the rheological properties of the mixture and 3D printing. They concluded not only that temperature and composition during 3D printing are responsible for the rheological properties of the mixture but also that the effect of material composition is more relevant than temperature. Table 2 shows that mixtures 2A and 2B have the highest consistency coefficient and the lowest *n* values. Fats play a key role in fine bakery products, not only because of the mouth feeling and nutritional needs of consumers but also to achieve the desired rheological properties with the aim of easier and more accurate printing [1].

Comparing the values of the consistency and apparent viscosities of all mixtures, mixtures 2 had the highest values. These mixtures contained rye flour, which has gluten, while flours in other doughs are gluten-free. Rye pentosans, proteins, and starch have an important function in water binding [19]. Rye flour contains 4–7% of pentosans, which greatly contribute to the viscosity of the dough [19]. Mixtures 2A, 2B, 3A, and 3B had higher apparent viscosity values than respective mixtures 1A and 1B, and this may be due to a higher fiber content [20]. Doughs 2 and 3 contained carob flour, which is known to have high dietary fiber content with large proportion of galactomannans [21]. The effect of fibers on dough rheology is related to their increased water-binding capacity [20]. For this reason, water amount was adjusted for mixtures 2A, 2B, 3A, and 3B until the appropriate rheological properties of the material, suitable for their printing, were achieved. Ultimately, those mixtures had 33.33% more water than 1A and 1B (Table 1). The consistency was generally lower (*p* < 0.001) for mixtures B than for mixtures A, probably because of water contained in the butter.

### 3.2. Dough Utilization Rate and Repeatability of Printing Weight

The average dough weight of individual printed sample (3.50 ± 0.18 g) between mixtures did not significantly differ, which is also shown as the total sum of all printed samples from the syringe at Figure 3a. Nonetheless, comparing the weight of the first and last printed 3D form, differences were significant (*p* < 0.001), regardless on the flour and fat type. The first samples of all six mixtures had the biggest weight compared with each subsequent printed form (Figure 3b). The weight of the first printed forms was 7–17% higher in relation to the masses of the last printed forms.

The percentage of mixture inserted into the printer syringe converted to final printed product is shown as dough utilization rate (Figure 3a). The highest mixture utilization rate (in average 79%) was recorded in mixtures A, i.e., those with olive oil (Figure 3a). Dough with olive oil is characterized by a better printability because of lubrication, increased dough elasticity, and better texture surface of the printed dough [1]. Test extrusion was performed before printing, which is the main reason for the loss of prepared mixtures during printing. In mixtures B, since carob flour contains harsher particles, the nozzle was occasionally clogged, and it was necessary to carry out additional test extrusion, which led to a lower utilization rate. In mixtures A, this difficulty did not occur due to contributions of added olive oil to cookie printing.

### 3.3. Physical Properties of Cookies

The weight of all shapes was reduced during baking and cooling resulting in baking loss (Table 3), probably due to the moisture loss. The correlation between baking loss and moisture content was significantly positive and moderate (r = 0.54). In agreement, Pulatsu et al. [11] reported 15–23% baking loss of different printed cookies differing in the type of fat (butter or shortening), sugar, and milk. The baking loss significantly depended on both flour and fat type, being lower for mixture 1 and for all A mixtures. The lower baking loss of mixture 1 could be attributed to the fact that it contained lower amount of water than the other mixtures. In addition, olive oil contained less water than butter. Higher baking loss of butter containing cookies can be also related to the fact that the dough with butter has a larger area of air–water contact than oil-containing dough and has a larger number of air bubbles incorporated, which are expected to form more channels during the baking process, resulting in greater water loss [22]. On the other hand, oil-containing dough has a lower ability to incorporate air bubbles during mixing compared to that with butter [10]. Sahi and Alava [23] studied the influence of emulsifiers on the rheological properties of dough and the incorporation of air bubbles into the dough during mixing and stated that the viscosity of the system is important for air retention. A higher viscosity value can retain air bubbles in the dough, which will lead to more channels through which water loss will occur, and at lower viscosity, air bubbles rise to the surface and are lost in the atmosphere [23]. In our work, we found an inverse correlation between baking loss and flow index (r = −0.81). Mixture 2B with the lowest flow index has the highest baking loss, while mixture 1A with the highest flow index has the lowest baking loss.

Table 3 shows the mean values of the color parameters L^*^, a^*^, b^*^, C, and H°. The expected significant difference (*p* < 0.05) of all color parameters was evident between mixtures 1 and 2, as well as 1 and 3 due to the carob flour added in mixtures 2 and 3. The lightness L^*^ and redness a^*^ were dependent on flour choice, where mixture 1 was the lightest, showing the lowest intensity of red color. The color parameters b^*^, C, and H° were significantly influenced by the interaction of flour and fat (*p* < 0.05), being the most intensive in sample 1A (oat with olive oil). Comparing the color of mixtures 1A and 1B (Table 3), the total difference was 2.81, which means that it was distinct. The color differences between samples 2A and 2B (Table 3) and 3A and 3B (Table 3) was 0.2 and 1.0, respectively, i.e., it was small. Since samples 1A and 1B contained light-colored oat flour, the difference in color caused by olive oil and butter was more pronounced. Carob flour, which gave dark color to samples of mixtures 2 and 3, masked the influence of olive oil on color. For all 3 mixtures, the H° values are in the range of 65.03–86.45, which indicates the visual impression of the red-orange color of the biscuits. The higher the C, the higher the color intensity of the sample that people perceive [24]. The C parameter has a higher value for mixture 1A compared to 1B, while for 2 and 3 mixtures, B have higher values than A.

### 3.4. Printing Repeatability and Accuracy

The biggest challenges of 3D food printing are the precision and accuracy of printing, its repeatability, and stability of 3D shapes. The printing quality and the stability of 3D shapes can be achieved not only by controlling the physicochemical, rheological, structural, and mechanical properties of the material [2] but also through extrusion conditions, i.e., extrusion rate, print speed, and nozzle diameter. Some of the printing quality parameters are the height and width of printed structure, which we measured at four positions of the printed form to examine the repeatability of printing (Figure 1). There were no significant differences in average values of studied parameters between two fillings of the syringe with the same mixture (*p* > 0.05).

Total height, line height, and width (Table 3 and Table 4 and Figure 4) were significantly influenced by the flour choice. Cookies from mixture 1 were the highest but also showed the widest printing line. No statistically significant differences were found between the average height and width of the 3D shape of mixtures A and B or between mixtures 2 and 3 (*p* > 0.05). The cookies from mixture 1 and 2 were higher when butter was used, but cookies from mixture 3 were higher when containing olive oil but the differences were small. The line width was positively correlated with height (r = 0.96) and inversely correlated with dough consistency (r = −0.77). Patterns of mixtures 2 and 3 were less thick and less spilled than patterns of mixtures 1, which indicates that carob flour, i.e., higher fiber content, increases the stability of mixtures.

The ideally printed layer had a height of 0.4 mm, and it was 2.6 mm wide then c = 0.154. To facilitate the presentation of the results, the c-value for the ideally printed layer was set to 1. For this reason, all calculated c-values for all samples were divided by 0.154. To ensure proper adhesion between the layers and to maintain the 3D-printed shape, the diameter of the printed line can be 130% of the nozzle diameter [17]. Since the diameter of the used nozzle was 2 mm, a “baseline” line width of 2.6 mm is acceptable. If c = 1, then the diameter of the printed line is equal to 130% of the nozzle diameter or 2.6 mm. If c < 1, then there was a spread of the printed line [24]. The c-value (Table 3) was significantly affected by the flour choice (Table 4). Base layer of mixture 2 spread the least, and the c-values were the closest to the value 1. Still, the significant difference was established only between samples 1B and 2A because also the fat type made small influence (*p* = 0.09). All doughs with olive oil had c-values that deviate less than 1 compared to those with butter (Table 3). The c-value of cookies was significantly correlated with dough apparent viscosity (r = 0.91) and consistency coefficient (r = 0.95). This is similar to previous study [8], which reported that the printability increases with a higher value of the consistency coefficient and the reduction in behavior index.

Nevertheless, each subsequent printed shape had lower line width depending on the flour choice (Figure 4). The height of subsequent printed samples was lowering only in printing mixture 1. Thus, the measured height and width of the printed line were significantly correlated with $\sqrt{\frac{Q}{v}}$ (r = 0.99 for both). At a higher extrusion rate, the deposition of the dough exceeds the right amount, which leads to the deposition of a wider printed layer of the mixture. This is because at a higher extrusion rate, a larger volume or mass of the mixture is extruded, whereby the diameter of the printed layer is larger than the diameter of the nozzle [25]. The first printed layer of the 3D shape of all mixtures was also the widest. At lower extrusion rates, more consistent printing with less or without spreading of the first printed layer was observed. Each subsequent layer applied to the “base” was narrower, i.e., equal to the diameter of the used nozzle.

At the value of $\sqrt{\frac{Q}{v}}$ 2.30, the width of the extruded line was close to or equal to 2.6 mm. At $\sqrt{\frac{Q}{v}}$ > 2.30, the nozzle speed was not sufficient to handle all the material extruded by the extruder, and excessive extrusion was observed, and thus the material was spilled. This case was observed in the first samples of all mixtures and to a lesser extent, in the last samples of mixtures 1A and 1B. Over extrusion of the material can also be confirmed by observing the height and width of the samples of all mixtures. The value of $\sqrt{\frac{Q}{v}}$ was dependent on flour type (Table 4), being the largest for mixtures 1A and 1B. It was inversely correlated with dough consistency coefficient (r = −0.70).

The average shape accuracy (97.5 ± 0.9%) was not significantly influenced with the flour or fat choice. Still, Figure 5 shows large deviations in the shape accuracy between the first sample and the last, ideal cookie form, significantly (*p* = 0.001) depending on both flour and fat type. Minor changes were observed in mixtures 2 and 3 with olive oil, which might be related to their rheological parameters, i.e., higher values of consistency coefficient and apparent viscosity. The increase in viscosity results in better shape retention during 3D food printing [11]. The shape accuracy was inversely correlated with the c-values of the mixtures (r = −0.82). Mixtures 2A and 2B, having c-values equal to or closest to 1, resulted in the highest printing accuracy (96.3–99.7%). Mixtures 3A and 3B show less desired shape precision (92.5–99.4%) and the least accurately printed were forms from mixtures 1A and 1B (94.2–98.3%). The largest sample weight was correlated (r = 0.69) with biggest deviation from the desired shape (Figure 3b). Although there was no significant difference (*p* > 0.05) in average weight between samples, significant (*p* < 0.001) differences were established in the weight of cookie samples along the printing time. The differences in weight, line width, and height at different printing times are the reasons for printing inaccuracy.

### 3.5. Dietary Fiber Content of Cookies

Whole grains contain insoluble and soluble dietary fiber with multiple health-promoting effects [26]. As expected, dietary fiber content of our cookies was substantially different depending only on the flour type (*p* < 0.001). Cookies from mixture 1 contained in average 2.02 ± 0.05 g of fiber per 100 of cookie dry matter. On the other hand, cookies from mixtures 2 and 3 were high in fiber containing in average 10.28 ± 0.28 and 8.31 ± 0.30 g of fiber per 100 of cookie dry matter, respectively. This confirmed the great potential of carob flour for the enrichment of cookies with fiber [27]. Besides, it enhanced nutritive value of cookies, and the fiber content influenced their technological properties; it inversely correlated with line width (r = −0.97), height (r = −0.96), and $\sqrt{\frac{Q}{v}}$ (r = −0.98). This is in agreement with previous findings that high-fiber raw materials added to pastes for 3D printing can enhance their processability, e.g., the rigidity of materials is raised after adding insoluble dietary fiber [8,28]. In cereal flours, insoluble fiber prevails over soluble fiber [26]. The carob flour used in our study was composed from both pulp and seeds. The carob seed endosperm contains high amount of galactomannans and is used for extraction of locust bean or carob gum [29]. Carob gum is used to gain higher baked product yields, to enhance dough viscosity and machinability, and to substitute gluten in bakery industry [29].

### 3.6. Principal Component Analysis

PCA extracted five factors with the average eigenvalue of 3.20, so only first two components above the average value with eigenvalues of 10.50 and 3.34, accounting for 86.5% of the total variance, were considered (Figure 6). The first component separates samples of mixture 1 from mixtures 2 and 3, while the second component separates samples of A mixtures from B mixtures (Figure 6a). A discrimination is clear between mixtures 1 and 2, but sample from mixture 3 are confused by other samples. The contributions of cases suggest that component 1 essentially contrasts sample 2A with 1A, whereas component 2 essentially contrasts sample 3B with samples 2A and 1A. The component 1 contributes highly to sample 1A but also 2A and 1B, while component 2 contributes most to the sample 3B (and additionally to 2A and 1A). The first component contrasts variables $\sqrt{\frac{Q}{v}}$, cookie color (L^*^, b^*^, C, and H°), and line width with the dough consistency, redness a^*^, c-value, and TDF (Figure 6b). The second component contrasts the baking loss with dough utilization rate and apparent viscosity. In particular, the sample 1A, which is characterized by the lowest fiber content, baking loss, and a^*^, but highest L^*^, b^*^, C, H°, line width, and $\sqrt{\frac{Q}{v}}$, is well separated from sample 2B.

## 4. Conclusions

The aim of this study was to determine the physical properties of 3D-printed whole grain cookies depending on flour and fat choice as well as on dough rheology. The results showed that the type of flour and the type of fat strongly affect dough rheology and printing accuracy. Mixtures containing more fiber have higher consistency coefficient, which is related with more narrow width of printed line and less spread of the printed layer resulting in better printing repeatability and accuracy. During 3D printing, each subsequent 3D cookie is printed at a lower extrusion rate, has a lower weight and more narrow width than the previous one, and is printed more accurately. The printing speed and extrusion rate are strongly related with the height and width of the printed dough layers. By modifying the cookie recipe and controlling the rheological properties of the dough, a cookie dough with higher nutritional value can be 3D printed with high accuracy and repeatability without the use of additives. Future studies should investigate the role of different sweeteners on the quality of 3D-printed cookies and test their acceptance by consumers.

## Figures and Tables

**Figure 1 foods-10-00193-f001:**
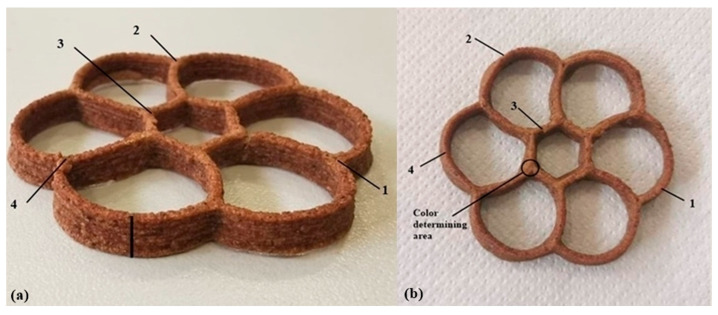
(**a**) Points for determination of shape height and (**b**) color determining area and 4 points for determining the width of the baked shapes.

**Figure 2 foods-10-00193-f002:**
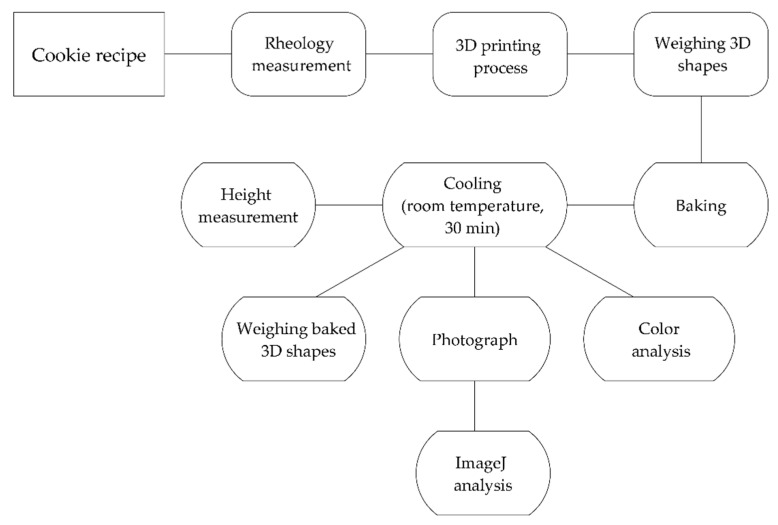
Schematic flow chart of the methods used in this study.

**Figure 3 foods-10-00193-f003:**
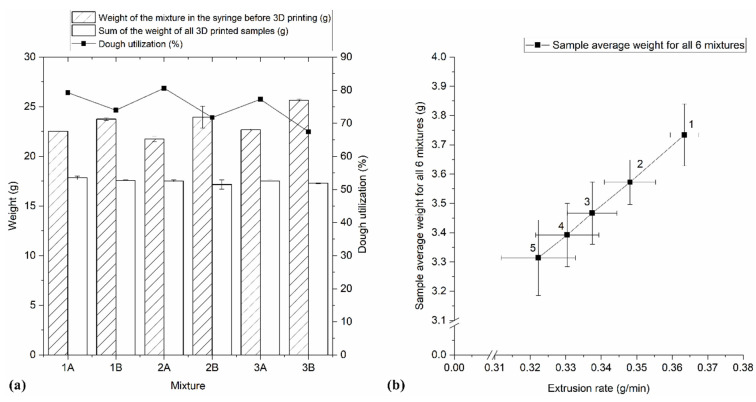
(**a**) Dough utilization rate of different mixtures in 3D printing and (**b**) average weight repeatability of 3D-printed cookies (in printing order 1–5) depending on the extrusion rate (mean ± standard deviation).

**Figure 4 foods-10-00193-f004:**
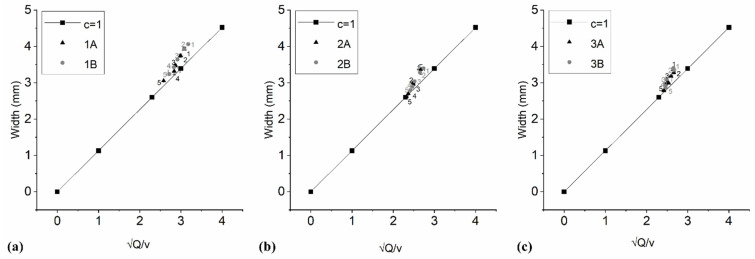
Line width of cookie from (**a**) mixture 1A-1B, (**b**) mixture 2A-2B, and (**c**) mixture 3A-3B as a function of extrusion rate (*Q*) and nozzle speed (*v*) for a syringe with a 2 mm diameter nozzle. Each point represents the average of 2 measurements with standard deviation represented as error bars.

**Figure 5 foods-10-00193-f005:**
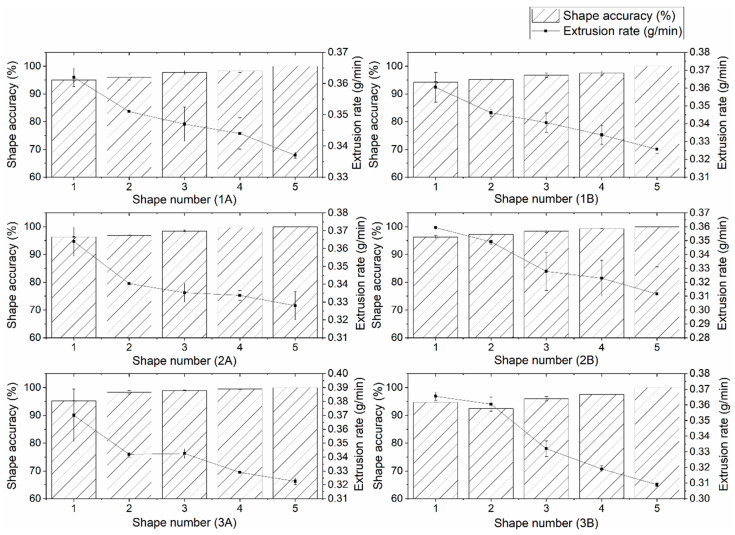
3D shape accuracy versus extrusion rate for each subsequent cookie sample (mean ± standard deviation, *n* = 2).

**Figure 6 foods-10-00193-f006:**
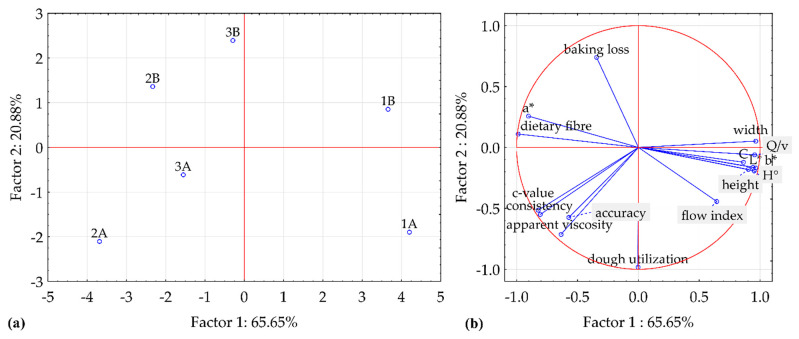
(**a**) The projection of samples and (**b**) the projection of responses on the factor-plane according to principal component analysis.

**Table 1 foods-10-00193-t001:** Mixtures and their composition prepared for 3D printing.

MIXTURES	Oat Flour (g)	Rye Flour (g)	Carob Flour (g)	Rice Flour (g)	Olive Oil (g)	Butter (g)	Water (mL)	Honey (g)
1A*	30	0	0	0	8	0	8	11
1B*	30	0	0	0	0	8	8	11
2A	10	10	10	0	8	0	12	11
2B	10	10	10	0	0	8	12	11
3A	10	0	10	10	10	0	12	11
3B	10	0	10	10	0	10	12	11

A*—olive oil; B*—butter.

**Table 2 foods-10-00193-t002:** Rheological parameters of the tested mixtures at 30 °C.

Mixture	Apparent Viscosity (Pas)	Consistency Coefficient (Pas^n^)	Flow Behavior Index
1A	63.2 ^c^	3.304 ^b^	0.659 ^a^
1B	25.38 ^e^	2.434 ^a^	0.282 ^bc^
2A	175.4 ^a^	16.559 ^e^	0.292 ^b^
2B	61.49 ^c^	7.182 ^c^	0.160 ^c^
3A	75.98 ^b^	9.498 ^d^	0.309 ^b^
3B	32.04 ^d^	3.284 ^b^	0.380 ^b^

^a–e^ Values within the same column marked with different letters significantly differ according to Tukey’s test (*p* < 0.05).

**Table 3 foods-10-00193-t003:** The appearance, baking loss, moisture content, color (L, a, b, C, H°), height, and line spreading (c-value) of cookies depending on the flour and fat type (mean ± standard deviation).

Sample	Appearance	Baking Loss (%)	Moisture Content (%)	L^*^	a^*^	b^*^	C	H°	Height (mm)	c-Value
1A	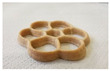	18.90 ^a^ ± 0.48	4.76 ^a^ ± 0.03	72.04 ^a^ ± 0.41	1.00 ^a^ ± 0.07	16.09 ^a^ ± 0.27	16.12 ^a^ ± 0.28	86.45 ^a^ ± 0.21	5.48 ^a^ ± 0.22	0.85 ^ab^ ± 0.05
1B	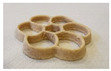	21.28 ^b^ ± 0.81	5.01 ^b^ ± 0.03	72.52 ^a^ ± 0.54	0.91 ^a^ ± 0.07	13.77 ^b^ ± 0.12	13.80 ^b^ ± 0.13	86.23 ^a^ ± 0.26	5.59 ^a^ ± 0.18	0.83 ^a^ ± 0.05
2A	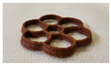	20.48 ^ab^ ± 0.44	5.29 ^c^ ± 0.02	49.88 ^b^ ± 0.26	5.02 ^b^ ± 0.15	10.75 ^e^ ± 0.28	11.85 ^c^ ± 0.32	65.03 ^c^ ± 0.15	4.91 ^a^ ± 0.09	0.91 ^b^ ± 0.08
2B	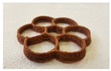	24.44 ^c^ ± 0.31	5.17 ^bc^ ± 0.05	50.07 ^b^ ± 0.84	5.22 ^b^± 0.33	11.32 ^de^ ± 0.64	12.46 ^bc^ ± 0.72	65.31 ^c^ ± 0.28	5.00 ^a^ ± 0.14	0.88 ^ab^ ± 0.06
3A	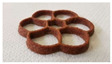	20.22 ^ab^ ± 0.26	5.14 ^bc^ ± 0.07	51.25 ^bc^ ± 0.30	5.63 ^b^ ± 0.09	12.18 ^cd^ ± 0.21	13.41 ^b^ ± 0.22	65.18 ^c^ ± 0.19	5.01 ^a^ ± 0.12	0.87 ^ab^ ± 0.06
3B	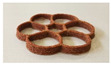	21.98 ^b^ ± 0.66	5.51 ^d^ ± 0.02	52.55 ^c^ ± 0.58	5.56 ^b^ ± 0.13	12.74 ^bc^ ± 0.33	13.90 ^b^ ± 0.35	66.40 ^b^ ± 0.30	4.87 ^a^ ± 0.20	0.84 ^ab^ ± 0.07

^a–e^ Values within the same column marked with different letters significantly differ according to Tukey’s test (*p* < 0.05).

**Table 4 foods-10-00193-t004:** ANOVA results for the influence of flour and fat on the properties of 3D shapes.

Predictor	Variable	Sum of Squares	*df*	Mean Square	*F*	*p*-Value
Flour	L^*^	5630	2	2815	1209	<0.001 *
Fat		6	1	6	3	0.119
Flour × Fat		3	2	2	1	0.485
Flour	a^*^	248.6	2	124.3	497.4	<0.001 *
Fat		0	1	0	0.1	0.821
Flour × Fat		0.3	2	0.1	0.5	0.605
Flour	b^*^	155.6	2	77.79	70.65	<0.001 *
Fat		2.3	1	2.33	2.11	0.152
Flour × Fat		27.8	2	13.9	12.62	<0.001 *
Flour	C	4.232	2	2.116	78.66	<0.001 *
Fat		78.62	2	39.31	29.65	<0.001 *
Flour × Fat		2.45	1	2.45	1.85	0.18
Flour	H°	27.56	2	13.78	10.39	<0.001 *
Fat		5779	2	2889	5225	<0.001 *
Flour × Fat		2	1	2	4	0.046 *
Flour	height	6	2	3	5	0.007 *
Fat		0.008	1	0.008	0.31	0.582
Flour × Fat		0.188	2	0.094	3.49	0.038 *
Flour	width	3.715	2	1.858	24.34	<0.001 *
Fat		0.184	1	0.184	2.41	0.126
Flour × Fat		0.059	2	0.029	0.39	0.682
Flour	weight	0.057	2	0.029	0.862	0.428
Fat		0.049	1	0.049	1.484	0.229
Flour × Fat		0.002	2	0.001	0.031	0.97
Flour	c-value	0.036	2	0.018	4.729	0.013 *
Fat		0.011	1	0.011	2.922	0.093
Flour × Fat		0	2	0	0.045	0.956
Flour	baking loss	45.6	2	22.8	8.73	0.001 *
Fat		127.3	1	127.3	48.68	<0.001 *
Flour × Fat		9.9	2	5	1.9	0.159
Flour	% shape accuracy	16.29	2	8.14	2.033	0.141
Fat		14.48	1	14.48	3.616	0.063
Flour × Fat		13.07	2	6.54	1.632	0.205
Flour	$\sqrt{\frac{Q}{v}}$	1.873	2	0.937	43.21	<0.001 *
Fat		0.032	1	0.032	1.46	0.233
Flour × Fat		0.04	2	0.02	0.92	0.405

* significant at *p* < 0.05.

## Data Availability

Data is contained within the article.

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
