# Peer review of "Influence of Flour and Fat Type on Dough Rheology and Technological Characteristics of 3D-Printed Cookies"

_foods, 2021, doi:10.3390/foods10010193_

Round 1

Reviewer 1 Report

This paper investigated the Influences of flour and fat type on dough rheology and technological characteristics of 3D printed cookies, which will be of interest for the readers of Foods. In my opinion, the study is original and novel. However, there are some flaws in the manuscript needed to be addressed.

Specific comments:

  1. L82-87: ~For experiments we used four types of flour: oat (72.1 g/100 g carbohydrates, 9.9 g/100 g proteins, 5.5 g/100 g lipids, and 2.3 g/100 g fiber) Eko – Jazo (Ivanovac, Croatia), rice (79.6 g/100 g carbohydrates, 7.2 g/100 g protein, 2.2 g/100 g lipids, and 2.9 g/100 g fibre), rye (77 g/100 g 84 carbohydrates, 9 g/100 g protein, 0.74 g/100 g lipids, and 11 g/100 g fibre) Garden (Zagreb, Croatia), and carob (81.4 g/100 g carbohydrates, 5.3 g/100 g protein and 0.5 g/100 g lipids, 26.3 g/100 g fiber) Perna (Vis, Croatia)~Q: Please rephrase the sentence.
  2. L97: ~ 2.3.D printing process ~Q: Please correct the typo.
  3. L98-99: ~The dough samples were fabricated using the extrusion-based 3D printer (Createbot 3D food printer-multi-ingredient support) with a syringe-extrusion-type plunger (volume of 20 mL).~Q: Please provide the proper information the 3D printer.
  4. L105-113: Q: The rheological measurement needs to be described more precisely. Why is the maximum shear rate different for the samples?
  5. L127-151: Q: The measurements of the line height and width are not properly addressed. How to calculate the accuracy? Please rephrase these sentences (L105-113).
  6. Table 2: Q: How to obtain the apparent viscosities of the samples in Table 2.

Author Response

We would like to thank the reviewers for the valuable comments on our manuscript. We have addressed all the comments. The revisions are highlighted using the "Track Changes" function in the manuscript text. Please find the details of our revisions and responses to the reviewers' comments bellow. Please detail the revisions that have been made, citing the line number and exact change.

REVIEWER 1

This paper investigated the Influences of flour and fat type on dough rheology and technological characteristics of 3D printed cookies, which will be of interest for the readers of Foods. In my opinion, the study is original and novel. However, there are some flaws in the manuscript needed to be addressed.

Thank you for the positive opinion on our manuscript. We have addressed all the issues and marked the changes in the text.

  1. L82-87: ~For experiments we used four types of flour: oat (72.1 g/100 g carbohydrates, 9.9 g/100 g proteins, 5.5 g/100 g lipids, and 2.3 g/100 g fiber) Eko – Jazo (Ivanovac, Croatia), rice (79.6 g/100 g carbohydrates, 7.2 g/100 g protein, 2.2 g/100 g lipids, and 2.9 g/100 g fibre), rye (77 g/100 g 84 carbohydrates, 9 g/100 g protein, 0.74 g/100 g lipids, and 11 g/100 g fibre) Garden (Zagreb, Croatia), and carob (81.4 g/100 g carbohydrates, 5.3 g/100 g protein and 0.5 g/100 g lipids, 26.3 g/100 g fiber) Perna (Vis, Croatia)~Q: Please rephrase the sentence.

A: We have rephrased the whole paragraph 2.1:

For experiments we used four types of flour, two types of fat and floral honey purchased at local grocery stores (Zagreb, Croatia). The used oat flour (Eko – Jazo, Croatia) contained 72.1 g/100 g carbohydrates, 9.9 g/100 g proteins, 5.5 g/100 g lipids, and 2.3 g/100 g fiber. Rice flour (Garden, Croatia) had 79.6 g/100 g carbohydrates, 7.2 g/100 g protein, 2.2 g/100 g lipids, and 2.9 g/100 g fibre. Rye flour (Garden, Croatia) was composed of 77 g/100 g carbohydrates, 9 g/100 g protein, 0.74 g/100 g lipids, and 11 g/100 g fibre. Carob flour (Perna, Vis, Croatia) contained 81.4 g/100 g carbohydrates, 5.3 g/100 g protein, 0.5 g/100 g lipids, and 26.3 g/100 g fiber. Butter contained 82 g/100 g lipids whereas olive oil had 91 g/100 g lipids.

  1. L97: ~ 2.3.D printing process ~Q: Please correct the typo.

A: Thank you, we have corrected the typo.

  1. L98-99: ~The dough samples were fabricated using the extrusion-based 3D printer (Createbot 3D food printer-multi-ingredient support) with a syringe-extrusion-type plunger (volume of 20 mL).~Q: Please provide the proper information the 3D printer.

A: We have added the required information on 3D printer: The dough samples were fabricated using the extrusion-based 3D printer (Createbot 3D food printer-multi-ingredient support, Ningbo Createbot Electronic Technology Co., Ltd. Zhejiang Province, China) with a syringe-extrusion-type plunger (volume of 20 mL) (lines 114-115).

  1. L105-113: Q: The rheological measurement needs to be described more precisely. Why is the maximum shear rate different for the samples?

A: The viscosity of all samples is now expressed at the same shear rate 5s-1 for better comparison (line 174). We have corrected the results in Table 2, Figure 6 and the correlation coefficient in paragraph 3.4 (line 345). Also, we have described better the methodology in 2.4.: The temperature of samples was maintained at 30±1˚C throughout the test using a Julabo F 33 refrigerated-heating circulator. The apparent viscosity (Pas) was measured at the shear rate 5 s-1. Also, the consistency coefficient (Pasn) and flow behavior index were calculated (lines 126-131).

  1. L127-151: Q: The measurements of the line height and width are not properly addressed. How to calculate the accuracy? Please rephrase these sentences (L105-113).

A: We have corrected the explanation on how height, width and accuracy were measured and calculated. The height was measured with digital micrometer whereas width and the shape accuracy were determined upon digital image analysis. We have provided a reference (16) for the determination of accuracy. Please find the following explanation in lines 154-176.

The total height of baked cookies was determined at 4 positions (Figure 1a) for each sample using a Digimet digital micrometer (Helios Preisser, Germany) (Figure 2). The total height was divided by the number of layers (twelve) to calculate the line height. The width of the printed layer was evaluated with digital image analysis using the ImageJ software (National Institutes of Health). All baked 3D forms (10 replicates) were photographed (with a Nikon D7500 (48 mm lens, ISO 450-500, 1/60; f/5.6) from the same fixed distance and images were calibrated on a real scale. The width of the initial printed layer was evaluated at 4 positions (Figure 1b, Figure 2) and is shown as mean value.

The shape accuracy of each printed form from the desired one was quantified by analysis of binary photographs (ImageJ, National Institute of Health) of the baked shapes [16]. The last printed 3D shape, given its dimensions (height and width) was the most accurately printed, and was set as the desired shape. Using the ratio of the number of black pixels to the total number of pixels in the photo, the shape accuracy of individual printed objects in relation to the desired 3D object was quantified and expressed in percentage.

  1. Table 2: Q: How to obtain the apparent viscosities of the samples in Table 2.

A: The apparent viscosity was determined according to ISO 2555:2018 (Determination of apparent viscosity by the brookfield test method) as stated in the paragraph 2.4.

Reviewer 2 Report

It was my pleasure to review paper titled: Influence of flour and fat type on dough rheology and technological characteristics of 3D printed cookies.

Overall, it's an interesting and well-prepared manuscript. The authors correctly described the aim, methods and obtained results. Statistical analysis of the results was also performed. I just have a few minor comments in the attached file.

Author Response

We would like to thank the reviewers for the valuable comments on our manuscript. We have addressed all the comments. The revisions are highlighted using the "Track Changes" function in the manuscript text. Please find the details of our revisions and responses to the reviewers' comments bellow. Please detail the revisions that have been made, citing the line number and exact change.

REVIEWER 2

Please add new keywords. The use of words previously used in the title of the manuscript should be avoided.

A: Thank you for the valuable comment. We have added and replaced some keywords: biscuits, extrusion rate, 3D printing precision and repeatability, dough consistency and viscosity, dietary fiber, carob flour, oat flour, olive oil

Line 42 This reference is quite old. Please add some newer positions, for example DOI: 10.1007/s00217-019-03370-5; 10.1016/j.jcs.2019.102879 - to mention a few

A: The previous reference (9) has been replaced with Kowalczewski, P.Ł., Walkowiak, K., Masewicz, Ł., Duda, A., Poliszko, N., Różańska, M.B., Jeżowski, P., Tomkowiak, A., Mildner-Szkudlarz, S., Baranowska, H.M.: Wheat bread enriched with raspberry and strawberry oilcakes: effects on proximate composition, texture and water properties. Eur. Food Res. Technol. 245, 2591–2600 (2019). https://doi.org/10.1007/s00217-019-03370-5

Also we have added the reference (28) Lille, M., Kortekangas, A., Heiniö, R. L., Sozer, N.: Structural and textural characteristics of 3D-printed protein-and dietary fibre-rich snacks made of milk powder and wholegrain rye flour. Foods9(11), 1527. (2020). https://doi.org/10.3390/foods9111527 for the discussion in paragraph 3.5.

Line 74 Please be more precise.

A: The aim of the study is now rephrased as (lines 80-85): The main aim of this research was to investigate the effect of butter substitution with olive oil and the influence whole grain and carob flour on the rheological properties of the dough and finally on 3D printing accuracy and repeatability. Also, the relationship between the printing process parameters and the rheological properties of the mixtures with the dimensions and accuracy of the 3D-printed forms was examined.

Line 91 and 93, 247, 256 Wrong way of citing.

A: Thank you, this is now corrected as: Lipton et al. [14], Pulatsu et al. [11], and Sahi and Alava [23] (lines 103, 105, 279, 289).

Line 97 2.3. 3D printing process

A: Thank you, the typo was corrected.

Line 107 mPas, mPa·s or mPa×s

A: It has been revised to mPas (line 123).

Figure 2 Graphics quality should be improved.

A: The figure graphics are now improved.

Line 192, 224, 230, 268, 272, 290, 294, 345, 346, 356 italic

A: The p is now formatted italic (lines 212, 219, 253, 260, 300, 304, 324, 328, 343, 371, 379, 380, 395).

Line 226 I am not sure if it makes sense to compare the yield of different doughs. The chemical composition of the raw materials varies greatly, resulting in significant differences. I suggest you delete this part.

A: We have changed the expression dough ‘yield’ to dough ‘utilization rate’, paragraph 3.2., which represents mass balance between the mixture inserted into the printer syringe and printed samples. We believe this might be usuful for future development of 3D printers and food products. We have also revised the description of its determination in paragraph 2.7: The utilization rate indicates what percentage of total mixture input in 3D printer entered into final product – cookie dough (lines 147-151).

Figure 3 Commas should be replaced with dots. Applies to all graphics. What are samples 1-6? Please do not change the markings of the samples.

A: Figure 3 and 6 are revised by replacing commas for dots. In figure 3, sample number (1-5) denotes the printing order. We have change figure caption as follows: Average weight repeatability of 3D printed cookies (in printing order 1-5) depending on the extrusion rate (mean ± standard deviation).

Round 2

Reviewer 1 Report

The authors have addressed the questions raised.

Q: The information provided for the oat, rice, rye and carob flours is confusing. Are Eko – Jazo, and Garden the names of brands or cultivars for the flours purchased?  

Author Response

Dear reviewer,

Eko – Jazo, and Garden are the names of brands of the flours purchased. Now it is written as:

The used oat flour (Eko – Jazo Ltd., Croatia) contained 72.1 g/100 g carbohydrates, 9.9 g/100 g proteins, 5.5 g/100 g lipids, and 2.3 g/100 g fiber. Rice flour (Garden Ltd., Croatia) had 79.6 g/100 g carbohydrates, 7.2 g/100 g protein, 2.2 g/100 g lipids, and 2.9 g/100 g fiber. Rye flour (Garden Ltd., Croatia) was composed of 77 g/100 g carbohydrates, 9 g/100 g protein, 0.74 g/100 g lipids, and 11 g/100 g fiber. Carob flour (Perna family-run farm, Vis, Croatia) contained 81.4 g/100 g carbohydrates, 5.3 g/100 g protein, 0.5 g/100 g lipids, and 26.3 g/100 g fiber. We hope that it is clear now.